# Augmented Concentration of Isopentyl-Deoxynyboquinone in Tumors Selectively Kills NAD(P)H Quinone Oxidoreductase 1-Positive Cancer Cells through Programmed Necrotic and Apoptotic Mechanisms

**DOI:** 10.3390/cancers15245844

**Published:** 2023-12-14

**Authors:** Jiangwei Wang, Xiaolin Su, Lingxiang Jiang, Matthew W. Boudreau, Lindsay E. Chatkewitz, Jessica A. Kilgore, Kashif Rafiq Zahid, Noelle S. Williams, Yaomin Chen, Shaohui Liu, Paul J. Hergenrother, Xiumei Huang

**Affiliations:** 1Department of Radiation Oncology, Indianapolis, IN 46202, USA; 2Department of Biochemistry and Molecular Biology, Indianapolis, IN 46202, USA; 3Department of Chemistry and Carl R. Woese Institute for Genomic Biology, University of Illinois at Urbana-Champaign, Urbana, IL 61801, USA; 4Department of Biochemistry, Simmons Comprehensive Cancer Center, UT Southwestern Medical Center, Dallas, TX 75390, USAnoelle.williams@utsouthwestern.edu (N.S.W.); 5Indiana University Health Pathology Laboratory, Indiana University School of Medicine, Indianapolis, IN 46202, USA; 6Eugene and Marilyn Glick Eye Institute, Department of Ophthalmology, Indiana University School of Medicine, Indianapolis, IN 46202, USA; 7Cancer Center at Illinois, University of Illinois at Urbana-Champaign, Urbana, IL 61801, USA; 8Indiana University Melvin and Bren Simon Comprehensive Cancer Center, Indiana University School of Medicine, Indianapolis, IN 46202, USA

**Keywords:** NQO1, IP-DNQ, ROS formation, DNA damage, programmed necrosis, apoptosis, pharmacokinetic

## Abstract

**Simple Summary:**

Although chemotherapy remains a fundamental treatment for a wide variety of human tumors, the emergence of resistance to chemotherapy presents a major hurdle in tumor therapy. Therefore, the identification of novel chemotherapeutic agents is essential to overcome this limitation and develop more effective tumor therapies. In this study, we illustrate that a novel NQO1 bioactivatable drug, IP-DNQ, effectively eradicates NQO1-positive cancer cells by inducing both apoptosis and programmed necrosis, displaying remarkable antitumor potential compared with previously tested NQO1 bioactivatable drugs. Mechanistically, IP-DNQ exterminates NQO1-positive cancer cells by generating excessive reactive oxygen species (ROS), thereby inducing DNA damage, PARP1 hyperactivation, and catastrophic energy loss. Overall, this study showcases the incredible antitumor efficacy of IP-DNQ against NQO1-positive cancer cells.

**Abstract:**

Lung and breast cancers rank as two of the most common and lethal tumors, accounting for a substantial number of cancer-related deaths worldwide. While the past two decades have witnessed promising progress in tumor therapy, developing targeted tumor therapies continues to pose a significant challenge. NAD(P)H quinone oxidoreductase 1 (NQO1), a two-electron reductase, has been reported as a promising therapeutic target across various solid tumors. β-Lapachone (β-Lap) and deoxynyboquinone (DNQ) are two NQO1 bioactivatable drugs that have demonstrated potent antitumor effects. However, their curative efficacy has been constrained by adverse effects and moderate lethality. To enhance the curative potential of NQO1 bioactivatable drugs, we developed a novel DNQ derivative termed isopentyl-deoxynyboquinone (IP-DNQ). Our study revealed that IP-DNQ treatment significantly increased reactive oxygen species generation, leading to double-strand break (DSB) formation, PARP1 hyperactivation, and catastrophic energy loss. Notably, we discovered that this novel drug induced both apoptosis and programmed necrosis events, which makes it entirely distinct from other NQO1 bioactivatable drugs. Furthermore, IP-DNQ monotherapy demonstrated significant antitumor efficacy and extended mice survival in A549 orthotopic xenograft models. Lastly, we identified that in mice IP-DNQ levels were significantly elevated in the plasma and tumor compared with IB-DNQ levels. This study provides novel preclinical evidence supporting IP-DNQ efficacy in *NQO1^+^* NSCLC and breast cancer cells.

## 1. Introduction

Cancer remains one of the most prevalent and life-threatening diseases, securing its position as the second leading cause of death worldwide, following cardiovascular disease. In 2020, lung cancer was responsible for 1.79 million deaths, while female breast cancer was diagnosed in approximately 2.26 million new cases, making them the leading cause of cancer-associated deaths and the most commonly diagnosed cancer, respectively [1,2]. Although chemotherapy and radiotherapy continue to be pivotal treatments for various cancer types, they also inflict damage upon normal and healthy cells. This collateral damage leads to undesirable side effects that significantly restrict their clinical use [3]. Over the past decade, targeted therapy has surfaced as a novel treatment approach and has been widely implemented in cancer patient care [4]. Nonetheless, only a limited number of targets have achieved clinical success. NAD(P)H quinone oxidoreductase-1 (NQO1), a xenobiotic-metabolizing enzyme, executes a two-electron oxidoreduction using NAD(P)H in the presence of various quinones [5]. NQO1 is upregulated in numerous solid tumors, including non-small-cell lung (NSCLC), breast, pancreatic ductal adenocarcinoma (PDAC), prostate, head and neck (HNC), and colorectal carcinoma, when compared with normal cells/tissues [6,7,8,9,10]. Consequently, NQO1 is identified as a potential target for cancer therapy. The development and clinical application of novel NQO1 bioactivatable drugs against cancer remain fervent areas of future research interest.

In previous studies, we reported on the effectiveness and limitations of certain NQO1 bioactivatable drugs such as β-lapachone (β-lap) [6,10,11,12,13,14,15,16,17] and deoxynyboquinone (DNQ) [18,19] (structures are presented in Appendix A). In the late 1980s, our laboratory commenced studies on β-lap (clinically formulated as ARQ761), a notable naphthoquinone due to its selective cytotoxicity towards NQO1-overexpressing cancer cells. β-Lap induces DNA damage, excessive PARP-1 activation, NAD^+^ and ATP depletion, and programmed cell necrosis [6]. However, the phase I clinical trial of ARQ761 revealed several treatment-related adverse events, including anemia, fatigue, hypoxia, nausea, and vomiting, with principal toxicities potentially related to methemoglobinemia [20]. To address these clinical challenges, our research group, along with other researchers, explored combination therapy, employing a sublethal dose of β-lap with ionizing radiation (IR) or other antitumor drugs (e.g., PARP1 inhibitors and PCNA inhibitors) [9,21]. We discovered that this combination therapy offers a feasible approach for selectively treating NQO1-overexpressing NSCLC [22], prostate cancer [8], and HNC [23]. Furthermore, the MTHFD2 blockade enhances the antitumor efficacy of β-lap with IR in HNC [24]. Additionally, the PCNA inhibitor T2 amino alcohol (T2AA) synergizes with β-lap to induce increased DNA damage, leading to programmed necrosis and G1 phase cell cycle arrest [21]; nicotinamide phosphoribosyl transferase (NAMPT) inhibition reduces the NAD^+^ pool in PDA cells, sensitizing them to ROS-mediated cell death by β-lap [25],and co-treatment suppresses SIRT1 activity, increasing the accumulation of acetylated FOXO1 in NSCLC [26]. Importantly, the combination of β-lap with Poly(ADP-ribose) Polymerase 1 (PARP1) inhibitors induces a cell death mechanism switch from PARP1 hyperactivation-mediated programmed necrosis to synergistic tumor-selective, caspase-dependent apoptosis [9]. Despite achieving success, to a degree, with combination therapies, the development of new NQO1 bioactivatable drugs remains an urgent necessity.

DNQ is another promising antineoplastic compound, sharing similar mechanisms with β-lap. DNQ undergoes an NQO1-dependent futile cycle, generating high levels of reactive oxygen species (ROS) that induce significant DNA damage, PARP hyperactivation, and NAD^+^/ATP depletion, leading to caspase-independent programmed necrosis [18]. Our previous study demonstrated that DNQ possesses a potency that is 10-fold greater than that of β-lap, significantly enhancing the therapeutic effect in in vitro human breast, prostate, and pancreatic cancer models. However, DNQ induces more severe side effects, such as high-grade methemoglobinemia at the low maximum tolerated dose (MTD) of 5 mg/kg in mice [18], limiting its clinical usefulness. Therefore, there is a critical need to develop novel DNQ derivatives that cause low-grade methemoglobinemia. In this study, we showed for the first time that a new drug, IP-DNQ, exerts high cytotoxicity towards NSCLC and breast cancer cells in an NQO1-dependent manner and that it elicited efficacious antitumor responses against mice bearing A549 orthotopic lung tumors.

## 2. Materials and Methods

### 2.1. Cell Lines and Culture

Endogenous NQO1-overexpressing human A549 non-small-cell lung cancer (NSCLC) cells, and MCF-7 and MDA-MB-231 breast cancer cells were obtained from the American Tissue Culture Collection (ATCC). A549 and MDA-MB-231 cell lines were manipulated to be deficient or proficient in NQO1, as previously described [21]. All cell lines were maintained in Dulbecco’s modified Eagle’s medium (DMEM) or RPMI 1640 medium containing 10% bovine calf serum. Cells were incubated in a humidified 95% air and 5% CO_2_ atmosphere at 37 °C. Culture media were replaced every other day and routinely screened for mycoplasma contamination. 

### 2.2. Antibodies

The following antibodies [21] were used for Western blotting and immunofluorescence included here: antibodies against NQO1 (1:200, sc-32793), β-actin (1:1000, sc-47778), PAR (1:200, sc-56198), α-tubulin (1:1000) and P53 (1:200, DO-1). These were obtained from Santa Cruz Biotechnology (Dallas, TX, USA). Anti-PARP1 (1:500, 4338-MC-50) was from Trevigen (Gaithersburg, MD, USA). Anti-γ-H2AX (1:1000, JBW301) was from Millipore (Burlington, MA, USA), and H2AX (1:1000, LS-C767192) antibody was from LSBio (Shirley, MA, USA).

### 2.3. Reagents and Chemicals

Isopentyl-deoxynyboquinine (IP-DNQ, 1-isopentyl-4,6-dimethylpyrido[3,2-g]quinoline-2,5,8,10(1H,9H)-tetraone) was synthesized and purified by Dr. Paul J Hergenrother’s laboratory in accordance with previous reports [27]. The synthesized compound was verified by NMR and high-resolution mass spectrometry, and purity was confirmed by LC-MS. IP-DNQ was dissolved in dimethyl sulfoxide (DMSO) at 1 mM for in vitro studies, it was dissolved in 20% of HPβCD at 12 or 15 mg/kg for in vivo studies, and then drug concentrations were determined with spectrophotometry, respectively. The rest of the reagents and chemicals [21], such as hydrogen peroxide (H_2_O_2_, HX0636) and dicoumarol (DIC, 287897), were purchased from Sigma-Aldrich (St. Louis, MO, USA).

### 2.4. Relative Survival Assays (DNA Assay)

A relative survival assay based on DNA content assessments was performed as previously described [21]. Briefly, 1 × 10^4^ cells were seeded per well in 48-well plates and treated with various doses of IP-DNQ (0.025–1 μM, 2 h) or with 0.2 μM IP-DNQ at indicated time point with or without competitive NQO1 inhibitor, DIC (50 μM). Cells were cultured further for approximately 5 days in drug-free medium until control cells reached ~90% confluence. After that, cells were washed with 1× PBS, and 250 μL of ddH_2_O was added and frozen at −80 °C overnight. The next day, the cells were freeze–thawed, lysed, treated, and assessed by a Victor X3 Plate reader (Perkinelmer, Waltham, MA, USA) as described previously [21]. All experiments were independently repeated 3 times. Statistical analyses methods were reported as means, and ±SD are described below.

### 2.5. O_2_ Consumption Rate (OCR) and Extracellular Acidification Rates (ECAR) Assessments

OCR and ECAR assessments were conducted using Seahorse 96-well dishes and sensor cartridges with a Seahorse XF96 analyzer. In brief, each well of Seahorse XF96 cell culture microplates was seeded with approximately 30,000 cells and then grown overnight. The following day, real-time OCR/ECAR measurements were taken using the analyzer according to the manufacturer’s instructions. The data were analyzed, and the results of each treatment group were graphed in at least three independent experiments.

### 2.6. Western Blotting

Total protein was extracted from cells treated with IP-DNQ using RIPA lysis buffer and quantified with bovine serum albumin standards along with the Bio-Rad protein assay. Different protein samples were separated by 7.0~12% SDS-PAGE, and then they were transferred to PVDF membranes and incubated with the corresponding antibody overnight [21,28]. Finally, immunoreactive bands were visualized using enhanced chemiluminescent detection, and each sample loading was measured and calculated with NIH ImageJ 2 software.

### 2.7. H_2_O_2_, ATP, and NAD^+^ Quantification

During or after treatments, H_2_O_2_, ATP, and NAD/NADH levels were tested at the indicated time points using respective standard chemical assays according to the manufacturer’s protocols (Promega G7572, Promega G8820, and Promega G9071) [21].

### 2.8. Immunofluorescence Assay

To assess the possibility of double-strand break (DSB) formation, cells were imaged with a Leica DM5500 fluorescence microscope to detect the presence of immunofluorescent foci of DSB formation. Background numbers of γ-H2AX foci were subtracted within each treatment experiment (50–100 per cell), and mean γH2AX ± SD intensity per cell was quantified and graphed.

### 2.9. Alkaline Comet Assay

The alkaline comet assay (Cell Biolabs STA-355) is a sensitive and relatively convenient technique for the detection of individual DNA lesions, including detecting DNA bases, single-strand breaks (SSBs), and DSBs. Glass slides were stained with SYBR green, and images were acquired using a Leica DM5500 fluorescence microscope. Comet tail lengths through alkaline comet assay were analyzed using NIH ImageJ. Each group was carried out at least 3 times and calculated as an average of 100 cells.

### 2.10. Caspase Activity Assay for Apoptosis Detection

A total of 48 h after treatment, A549 or MDA-MB-231 *NQO1^+^* cells were assessed for caspase 3/7 enzymatic activities following the manufacturer’s protocols (Promega G8091; Madison, WI, USA).

### 2.11. Annexin-V FITC/PI Assay

The assay was specifically designed for using a short and one-step staining method to detect apoptosis. After treatment with drugs, cells were harvested, and then 1 × 10^6^ cells were resuspended in staining buffer after washing with 1X PBS. All cells were stained with a combination of Annexin V and propidium iodide (PI) dye for 10 min according to the protocol. An amount of 400 μL of flow cytometry staining buffer was added to run flow cytometry, and ultimately, each point apoptosis event was analyzed with the FlowJo 10 software.

### 2.12. In Vivo Antitumor Efficacy and Pharmacokinetic (PK) and Pharmacodynamic (PD) Studies

Antitumor, survival, pharmacokinetic (PK), and pharmacodynamic (PD) studies were performed using orthotopic NSCLC A549-luc xenograft-bearing NSG mice. Bioluminescence (BLI)-based tumor volumes, long-term survival, and target validation assays were performed with log-rank tests for survival [21]. PK of IP-DNQ/IB-DNQ levels in blood and tumor were assessed using LC-MS/MS analyses following the extraction of plasma or tumor homogenates with acetonitrile [9]. PD studies of NAD^+^ and ATP levels in tumors were performed [9]. All animal experiments were carried out under a protocol approved by the Indiana University School of Medicine Institutional Animal Care and Use Committee (IACUC), following guidelines for the ethical conduct in the handling and care of animals used for research.

### 2.13. Statistical Analysis

All experiments were carried out more frequently than three biological replicates, and data are presented as mean ± standard deviation (SD). ANOVA or two-tailed Student’s *t*-test was used for multiple comparisons via statistical analysis. Values of *p <* 0.05 were deemed to be statistically significant. * *p <* 0.05, ** *p <* 0.01, and *** *p <* 0.001.

## 3. Results

### 3.1. IP-DNQ Exerts Potent Cytotoxicity in Various Types of NQO1^+^ Cancer Cells

To compare the cytotoxicity levels of β-lap, DNQ, and IP-DNQ, we first assessed their lethality in two endogenous NQO1-expressing cancer cell lines: MCF-7 (breast) and A549 (NSCLC) (Figure 1A,B). Both cell lines were treated with various concentrations of β-lap (1–20 µM), DNQ (0.025–1 µM), and IP-DNQ (0.025–1 µM), with or without dicoumarol (DIC, 50 µM), an NQO1-specific inhibitor, for two hours. IP-DNQ exhibited potent cytotoxicity, with IC_50_ values of 0.025 µM in the MCF-7 cells and 0.08 µM in the A549 cells, similar to the parent drug, DNQ. DIC significantly suppressed lethality in both cell lines (Figure 1A,B). Interestingly, the MCF-7 cells were hypersensitive to all three drugs (IC_50_ 0.08 µM for β-lap treatment), and IP-DNQ was approximately 30-fold more potent than β-lap, with an IC_50_ of 2.5 µM in the A549 cells (Figure 1A,B). The cytotoxicity of IP-DNQ in the A549 cells was confirmed by a colony formation assay (Appendix A).

Subsequently, we utilized CRISPR/Cas9-mediated *NQO1* knockout A549 cells to test the cytotoxic effects of IP-DNQ. As anticipated, the A549-*NQO1*-KO cells remained resistant to IP-DNQ treatment (Figure 1C). MDA-MB-231 breast cancer cells, which lack NQO1 expression due to a *2 polymorphism, were found to be resistant to β-lap [9] and DNQ [18]. Upon reconstituting NQO1 in the MDA-MB-231 cells, we observed that these cells became selectively hypersensitive to IP-DNQ (Figure 1D). The knockout or reconstitution of NQO1 in A549 and MDA-MB-231 cells was confirmed by Western blotting, and the NQO1 enzyme activity of all of the aforementioned cells was determined by an ELISA analysis (Appendix A).

Furthermore, we exposed MCF-7, A549, MDA-MB-231 *NQO1^+^*, and A549-*NQO1*-KO cells to 0.2 µM of IP-DNQ for a duration of 120 min to determine the minimum time to reach cell death. As expected, the MCF-7 cells were the most sensitive to IP-DNQ, and 20 min of exposure to IP-DNQ resulted in 60% cell death compared to approximately 20–30% of cell death in the other three cell lines (Figure 1E–H). The minimum time to kill >90% of MCF-7 cells was 40 min, compared to approximately 80 min for the other three cells (Figure 1E–H). Altogether, these data suggest that IP-DNQ efficiently kills breast and NSCLC cells in an NQO1-dependent manner.

### 3.2. IP-DNQ Induces NQO1-Dependent ROS Formation, PARP1 Hyperactivation, and NAD^+^/ATP Loss

Previous studies have reported that β-lap and DNQ undergo a futile redox cycle mediated by NQO1, leading to the formation of reactive oxygen species (ROS), the hyperactivation of poly(ADP-ribose) polymerase-1 (PARP-1), and NAD^+^/ATP depletion [6,9,18]. Given that IP-DNQ is a DNQ derivative, we investigated whether IP-DNQ kills cancer cells via similar mechanisms. We first examined the Oxygen Consumption Rates (OCRs) in A549 cells treated with IP-DNQ (0.1 or 0.25 µM) with or without dicoumarol (DIC). The OCR levels were monitored with a Seahorse analysis at 9 min intervals for 200 min. We observed that a sublethal IP-DNQ dose (0.1 µM) achieved an oxygen consumption of around 120 pmol/min between 30 and 70 min, which gradually decreased until 200 min passed (Figure 2A). A lethal IP-DNQ dose (0.25 µM) dramatically increased the OCR levels to 200 pmol/min, maintaining >150 pmol/min between 30 and 70 min, before finally decreasing to the lowest level at 150 min (Figure 2A). Additionally, the extracellular acidification rates (ECAR) were also monitored (Figure 2B). Since oligomycin inhibits ATP synthase, the cells’ glycolytic reserve capacities significantly increased after its addition [12]. However, the IP-DNQ treated cells reversed this capacity in a dose-dependent manner (Figure 2B), with the DIC blocking the OCRs/ECAR induced by IP-DNQ (Figure 2A,B).

We then examined whether IP-DNQ affected ROS formation. In a previous study, we found that 0.25 µM of DNQ time-dependently increased ROS formation, with the drug treatment groups exhibiting levels that were 7-fold higher than the control groups at 60 min [18]. A549 and A549-*NQO1*-KO cells were treated with or without IP-DNQ ± DIC at the indicated doses, and two hours later, the ROS levels were measured by the H_2_O_2_ levels. Interestingly, the H_2_O_2_ levels in the 0.25 µM IP-DNQ treated cells were approximately 50-fold higher than those in the control group (Figure 2C). In contrast, IP-DNQ-induced ROS generation was reversed by either DIC-treated or *NQO1*-knockout cells (Figure 2C). Furthermore, a Western blot analysis indicated that treatment with 0.25 µM of IP-DNQ led to an increase in PAR formation at 15 and 30 min, followed by a decrease from 60 to 120 min, which was concurrent with the appearance of the γ-H2AX protein, a surrogate for DNA double-strand break formation (Figure 2D). The exposure of A549 cells to sublethal (0.02–0.1 µM) and lethal (0.2 µM) IP-DNQ doses for 5 min resulted in the induction of increasing DNA damage, leading to a corresponding increase in PAR formation (Figure 2E and Appendix A). However, at super-lethal IP-DNQ doses (0.4, 0.6, and 0.8 µM), NAD^+^ consumption was rapid and exceeded the available supply within 5 min, causing PAR formation to exhibit a decreasing trend under these conditions (Appendix A). Conversely, neither the PAR nor γ-H2AX protein was noted in the A549 *NQO1*-knockout cells after IP-DNQ treatment, indicating that IP-DNQ kills cancer cells in an NQO1-dependent manner (Figure 2D). Similar changes in PAR and γ-H2AX protein levels post-IP-DNQ treatment were confirmed in MCF-7 cells (Figure 2F). Furthermore, IP-DNQ induced NAD^+^/ATP loss in dose- and time-dependent manners, while DIC or *NQO1* knockout rescued the cells from IP-DNQ-induced catastrophic NAD^+^/ATP energy loss (Figure 2G–L). Altogether, our findings suggest that IP-DNQ induces substantial ROS production and DNA damage, resulting in the hyperactivation of PARP activity. Subsequently, the PAR levels dramatically decrease due to NAD^+^/ATP pool exhaustion.

### 3.3. IP-DNQ Induces NQO1-Dependent DNA Damage, and Ca^2+^ Plays a Pivotal Role in IP-DNQ-Induced Cytotoxicity

DNA damage has been identified as a consequence of cell stress induced by ROS [29]. Our previous data demonstrated that IP-DNQ induced foci formation, as evidenced by a Western blot analysis (Figure 2D,F). In this context, we utilized the immunostaining of γ-H2AX foci formation/nucleus to corroborate these results. As anticipated, the exposure of A549 cells to the lethal dose of 0.25 µM of IP-DNQ led to a progressive increase in foci formation (~15–30) from 30 min to 120 min (Figure 3A,B). In contrast, DIC mitigated the IP-DNQ-mediated foci formation (Figure 3A,B). We also examined the total DNA damage (e.g., base damage, SSBs, and DSBs) following IP-DNQ treatment. The A549 cells treated with a sublethal dose of 0.1 µM of IP-DNQ barely displayed an increment in the comet tail length, while 0.25 µM of IP-DNQ significantly induced notable changes in the comet tail length to 65 ± 10 a.u. versus the control, which was 6 ± 2 a.u. (Figure 3C,D). Conversely, DIC suppressed the treatment-induced increase in the comet tail length (Figure 3C,D).

Ca^2+^ has been demonstrated to be a vital co-factor in PARP-1 hyperactivation following ROS-induced DNA damage, subsequently altering cellular metabolism and DNA repair [30]. To ascertain its role during IP-DNQ exposure, we utilized BAPTA-AM, an endogenous cytosolic Ca^2+^ chelator. A549 cells were pretreated with or without 5 µM BAPTA-AM for 1 h and co-treated with various doses of IP-DNQ for 2 h, followed by a DNA assay. BAPTA-AM treatment markedly enhanced the cell survival rate following IP-DNQ treatment (Figure 4A). Moreover, we discovered that BAPTA-AM dramatically obstructed PAR formation induced by IP-DNQ (Figure 4B) and prevented NAD^+^/ATP losses at any indicated time, even at the lethal dose of 0.25 µM of IP-DNQ treatment (Figure 4C,D). In summary, the immunofluorescence and comet data demonstrated that IP-DNQ triggered DNA damage dependent on NQO1, and Ca^2+^ was pivotal in the hyperactivation of PARP1 induced by IP-DNQ.

### 3.4. IP-DNQ Initiates Both Apoptosis and Programmed Necrosis in NQO1^+^ Cancer Cells

Treatment of *NQO1*^+^ cancer cells with a lethal dose of β-lap or DNQ alone induces PARP1 hyperactivation-mediated programmed necrosis (NAD^+^-Keresis). However, when PARP1 inhibitors are combined with β-lap, they induce tumor-selective, caspase-dependent apoptosis [9,18]. In this study, we utilized annexin-V and PI double staining to examine cell death. A549 cells were exposed to either 0.1 or 0.25 µM of IP-DNQ ± DIC for 2 h. After a duration of 48 h, the cells from all of the tested groups were stained using annexin-V and PI, and subsequently analyzed using flow cytometry. The A549 cells that were treated with staurosporine (STS) (1 µM) for 18 h served as a positive control for apoptosis. IP-DNQ significantly increased both early apoptotic cells (Q3) and the total cell death (Q1 + Q2 + Q3) in a dose-dependent manner, whereas DIC reduced cell death (Figure 5A).

To identify the specific cell death pathway triggered by IP-DNQ, we treated MCF-7 cells with IP-DNQ (0.25 μM) ± ZVAD-fmk (pan-caspase inhibitor, 75 μM) ± DIC (50 μM) for 2 h. After 24 hour treatment, we extracted the total protein and identified the presence of PARP1 and p53 through a Western blot analysis. Additionally, the cells were subjected to STS (1 μM) for 18 h, either with or without ZVAD-fmk (75 μM), serving as a positive control for apoptosis. Treating MCF-7 cells with a lethal dose of IP-DNQ alone (0.25 μM) resulted in an atypical proteolytic cleavage of PARP1 (60 kDa) and p53 (40 kDa), suggesting that cell death was activated by IP-DNQ via a programmed necrosis event (Figure 5B, lane 2). Concurrently, this sample exhibited an 89 kDa band consistent with PARP1 cleavage-mediated proteolysis, indicating the presence of an apoptotic pathway upon the exposure to IP-DNQ (Figure 5B, lane 2). Furthermore, the STS positive control group exhibited the 89 kDa PARP1 cleavage band, which implies that the MCF-7 cells underwent apoptosis when treated with STS. Notably, ZVAD-fmk prevented the 89 kDa PARP1 cleavage in the STS group (Figure 5B, lane 7&8), but it did not inhibit the 60 kDa PARP1 and 40 kDa p53 cleavages induced by IP-DNQ (Figure 5B, lane 3). We also treated MDA-MB-231 *NQO1^+^* and A549 cells with IP-DNQ (0.25 μM), with or without ZVAD-fmk, to examine the caspase 3/7 enzymatic activity. IP-DNQ treatment led to a significant increase in the caspase 3/7 enzyme activity in both cell lines, while ZVAD-fmk effectively neutralized these effects (Figure 5C). In conclusion, our findings indicate that IP-DNQ triggers both apoptosis and programmed necrosis in NQO1-positive cancer cells. This suggests that IP-DNQ might utilize diverse mechanisms to eliminate cancer cells compared to its parent drug, DNQ.

### 3.5. Antitumor Efficacy of IP-DNQ in Orthotopic NSCLC Xenografts, Pharmacokinetics (PK) and Tissue Pharmacodynamics (PD)

After obtaining compelling evidence of the antitumor effects of IP-DNQ in vitro, we aimed to validate our findings in vivo. We examined the antitumor efficacy of IP-DNQ in NQO1-expressing A549 cells using orthotopic xenografts. Female NSG mice (18–20 g) were injected with ~1 × 10^6^ A549 or 1.3 × 10^6^ A549-*NQO1*-KO cells via the tail vein. Seven days later, the mice (n = 5/group) were treated with vehicle (HPβCD, iv) or HPβCD-IP-DNQ (8 or 12 mg/kg, iv) every other day for a total of five treatments. We used bioluminescence imaging (BLI) to monitor the relative tumor volumes on days 18, 26, 46, 67, and 90 (Figure 6A). The quantitative analyses of the BLI intensities revealed that the vehicle (HPβCD) treatment dramatically increased the tumor volume, while HPβCD-IP-DNQ (8 or 12 mg/kg) significantly suppressed lung tumor growth in a dose-dependent manner (Figure 6B). The Kaplan–Meier survival curves demonstrated that both the 8 and 12 mg/kg HPβCD-IP-DNQ treatments significantly extended the mice’s lifespans, and notably, the mice that were treated with the higher dose of HPβCD-IP-DNQ (12 mg/kg) lived for more than 90 days (Figure 6C). Additionally, we utilized the A549 control and A549-*NQO1*-KO cells to establish orthotopic lung xenografts simultaneously. Because the knockout of NQO1 in the A549 cells inhibited cell proliferation in vitro (Appendix A), the inoculation of 1 × 10^6^ A549 cell or 1.3 × 10^6^ A549-*NQO1*-KO cells resulted in similar rates of tumor growth in vivo (Appendix A). Therefore, for Figure 6D, these compensatory inoculations were used. *NQO1* knockout entirely negated the antitumor efficacy of IP-DNQ, highlighting the NQO1-dependent effect of IP-DNQ in vivo (Figure 6D). In Figure 6E–H, the mice (n = 3/group) received only one injection and were sacrificed at the indicated time interval (5–120 min). As shown, the IP-DNQ treatment resulted in a substantial reduction in the NAD^+^ and ATP levels, which is consistent with prior in vitro observations (Figure 6E). The pharmacokinetic analyses showed that the serum/tumor concentrations of IP-DNQ were detectable throughout the 120 min study, with a half-life comparable to that of DNQ/IB-DNQ (Figure 6F). However, the IP-DNQ levels were significantly higher in the plasma and tumors compared to the IB-DNQ levels (Figure 6G–H). IB-DNQ, one of the DNQ derivatives, is an established anticancer agent without immediate or delayed hematologic, non-hematologic, or off-target oxidative toxicities [31].

To compare the antitumor efficacy of β-lap, DNQ, IB-DNQ, and IP-DNQ, A549 orthotopic tumor-bearing mice were treated with the maximum tolerated doses (MTDs) of β-lap (25 mg/kg), DNQ (5 mg/kg), IB-DNQ (15 mg/kg), or IP-DNQ (15 mg/kg), or with the vehicle. Relative to the vehicle, DNQ and β-lap administrations individually led to approximately 15–19% and 10% losses in the mice’s body weights, respectively. In contrast, the IB-DNQ and IP-DNQ treatments resulted in less than a 5% loss in the mice’s body weights, with these groups recovering their weights faster, reaching the levels equivalent to the vehicle-treated group (Figure 6I). The Kaplan–Meier survival curves indicated that the DNQ and β-lap treatments extended the mice’s lifespans to around 60 and 70 days, respectively, while IB-DNQ and IP-DNQ extended them to over 100 and 110 days (Figure 6J). Furthermore, no long-term normal tissue (liver) toxicities were observed after the administration of these four drugs (Figure 6K). To further confirm the impact of NQO1 bioactivatable drugs on liver function, the plasma ALT (Alanine Transaminase) and AST (Aspartate Aminotransferase) were measured following the drug treatments. IP-DNQ and IB-DNQ induced a mild elevation in the ALT and AST enzyme activities, while DNQ and β-lap significantly increased both the ALT and AST enzyme activities. Particularly, with the DNQ treatment, even after 72 h post-treatment, these enzymes did not completely return to the baseline levels (Appendix A). During the five dosing sessions, we monitored for signs of methemoglobinemia. IP-DNQ and IB-DNQ treatments resulted in milder side effects (such as no ruffled fur, no weight loss, less hunching, and reduced tachypnea and jumping) compared to the β-lap and DNQ groups (Appendix A). Overall, our findings suggest that IP-DNQ effectively targets human NQO1^+^ tumors with fewer side effects (specifically, methemoglobinemia) in vivo.

## 4. Discussion

For the past three decades, the global medical community has concentrated its efforts on advancing cancer treatment. A significant number of these patients grapple with lung and breast cancers. The breakdown of lung cancer subtypes reveals non-small-cell lung cancer (NSCLC) to be predominant, accounting for about 85% of cases. Statistics from the American Cancer Society in 2021 highlighted a sobering forecast: there were an estimated 131,880 fatalities and 235,760 novel lung cancer diagnoses in the USA. Parallelly, the prognosis for breast cancer was equally concerning, with an expected 281,550 new diagnoses and roughly 43,600 mortalities [32].

Given that lung and breast cancers consistently rank as the most prevalent cancer types, there is an unequivocal demand for more effective treatment modalities. Conventional chemotherapy, while effective, does not discriminate and targets both malignant and benign replicating cells. This lack of selectivity often results in damage to healthy tissues. Considering these limitations, the therapeutic landscape is moving towards targeted therapies that promise precision. However, a significant challenge is that many of these “targeted” agents are not adequately selective. The early 1960s witnessed the discovery of DT-diaphorase (later renamed NQO1) by Ernster and team, leading to extensive studies on its role in various cancers [33]. Ongoing research underscored the overexpression of NQO1 in several aggressive cancers, with its heightened levels being inversely related to survival rates, especially in NSCLC, breast, and colon cancers [34,35,36].

For the past two decades, our laboratory’s focus has largely been on understanding the mechanism of NQO1-bioactivatable agents, particularly β-lap [6,37,38,39] and DNQ [18]. Although both showed promise, they also exhibited significant toxicity in animal models. Specifically, the clinical trials of β-lap identified serious methemoglobin-related problems, despite its impressive antitumor efficiency [20]. Addressing these issues, our team developed IP-DNQ, a DNQ derivative, which remarkably reduces acute toxicity in mice without sacrificing its efficacy. The primary objective of this study was to thoroughly assess the antitumor efficacy of IP-DNQ. Our investigation covered both in vitro and in vivo environments, conclusively highlighting the power of IP-DNQ, especially against NQO1-expressing cells. Our results accentuated the crucial role of NQO1 in the antitumor effect of IP-DNQ. The significant difference in antitumor efficacy between the A549 control and A549-*NQO1*-KO cells clarifies the NQO1-dependent action of IP-DNQ. Such findings may guide patient selection in clinical settings, ensuring that the drug is administered to those most likely to benefit. Recognizing the role of NQO1 could also lead to the development of combination therapies that enhance NQO1 expression or activity, thereby potentially boosting the therapeutic impact of drug.

Another key element of our research was comparing IP-DNQ with other potential antitumor agents like β-lap, DNQ, and IB-DNQ. Our previous studies have demonstrated that the β-lap and DNQ treatments induce methemoglobinemia [40,41,42]. Although these four compounds possess unique antitumor abilities, IP-DNQ stood out due to its extended effect on mice’s lifespans and fewer side effects, especially concerning methemoglobinemia. This aspect is crucial because side effects often restrict the therapeutic range of many anticancer agents, making IP-DNQ’s reduced side effect profile a considerable advantage for potential clinical applications. Methemoglobinemia results from the oxidation of hemoglobin, transforming its heme iron from a ferrous (Fe^2+^) state to a ferric (Fe^3+^) state, which prevents it from binding to oxygen and subsequently impairs oxygen delivery to tissues [43]. The mild methemoglobinemia caused by IP-DNQ might stem from adding the isopentyl group to the DNQ precursor. This change results in reduced hemoglobin oxidation, which can still bind to and transport oxygen to tissues.

In summary, our study presents encouraging findings about IP-DNQ’s potential as an effective antitumor agent, especially for NQO1-expressing NSCLC cells. The drug’s prominent antitumor effects, combined with its minimized side effect profile, mark it as a candidate deserving further clinical evaluation. As oncology continues to search for treatments balancing efficacy with safety, IP-DNQ emerges as a promising contender. Future studies should consider exploring combination therapies and diving deeper into the molecular mechanisms underpinning the antitumor actions of IP-DNQ.

## 5. Conclusions

In conclusion, by thoroughly elucidating the molecular mechanism of IP-DNQ against NQO1-positive cancers, our study offers the first translational evidence supporting the exploration of combination therapy with IP-DNQ and ionizing radiation (IR) or other antitumor drugs (e.g., PARP1 inhibitors and PCNA inhibitors).

## Figures and Tables

**Figure 1 cancers-15-05844-f001:**
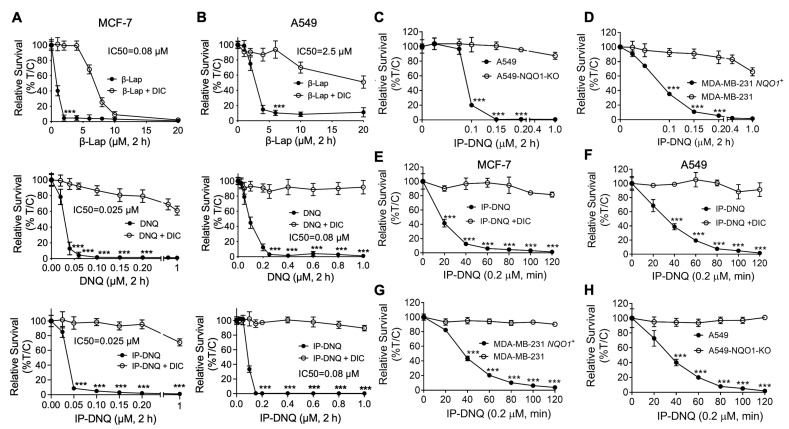
**IP-DNQ induces potent cytotoxicity in various NQO1-overexpressing cancer cells.** (**A**–**D**) Breast cancer cell line MCF-7 (**A**), NSCLC cell line A549 (**B**), breast cancer cell line MDA-MB-231 WT and NQO1-overexpressing MDA-MB-231 *NQO1^+^* cells (**C**)**,** and A549-*NQO1*-KO cells (**D**) were treated with various doses of β-lap (0–20 µM), DNQ (0–1.0 µM), or IP-DNQ (0–1.0 µM) ± DIC (dicoumarol, 50 µM, an NQO1 inhibitor) for 2 h. After this, drugs were removed, and cell viability was determined with a relative survival assay (DNA assay) 7 days later. (**E**–**H**) All of the aforementioned cell lines were treated with IP-DNQ (0.2 µM) at the indicated time points, and long-term relative survival assays were performed. Each group has six replicates for every biological repeat. Results (mean ± SD) in (**A**–**H**) represent experiments performed at least three times. *** *p* < 0.001 (*t* tests).

**Figure 2 cancers-15-05844-f002:**
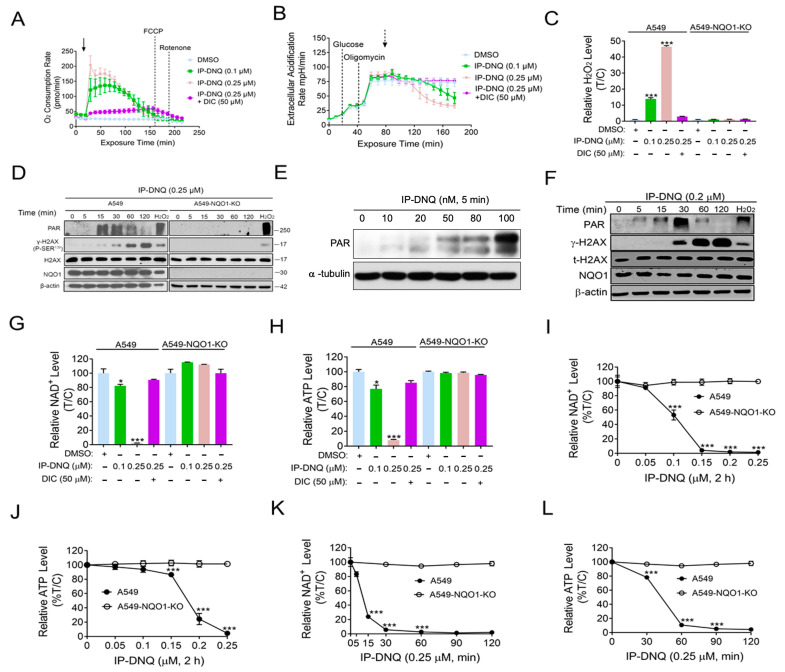
**IP-DNQ leads to ROS formation, PARP1 hyperactivation, and NAD^+^/ATP depletion.** (**A**,**B**) A549 cells were treated with DMSO or IP-DNQ (0.1 or 0.25 µM) ± DIC (50 µM) for 2 h. Real-time oxygen consumption rates (OCRs) and extracellular acidification rates (ECARs) after various drug treatments (added at t = 20 min, solid arrow) were monitored at 9 min intervals for 200 min. Dashed arrow indicates the addition of 2-DG at 80 min. (**C**) A549 and A549-*NQO1*-KO cells were treated with DMSO or IP-DNQ (0.1 or 0.25 µM) ± DIC (50 µM) for 2 h. ROS formation was assessed using H_2_O_2_ level kit. (**D**) Same cells as Figure 2C were treated with DMSO or IP-DNQ (0.25 µM), and cell extracts were prepared at the indicated time points. Cells were also exposed to H_2_O_2_ (500 µM, 15 min in PBS) as a positive control. Samples were assessed for PAR formation, γ-H2AX, total H2AX, NQO1, and β-actin (loading control). (**E**) A549 cells were exposed to DMSO and various dosages of IP-DNQ (0.02–0.1 µM) for 5 min. Samples were assessed for PAR formation and β-actin (loading control). (**F**) MCF-7 cells were treated with DMSO or IP-DNQ (0.2 µM), and cell extracts were prepared at the indicated time points. Cells were also exposed to H_2_O_2_ (500 µM, 15 min in PBS) as a positive control. Samples were assessed for PAR formation, γ-H2AX, total H2AX, NQO1, and β-actin (loading control). (**G**,**H**) The same cells as those in Figure 2C were treated with DMSO or IP-DNQ (0.1 or 0.25 µM) ± DIC (50 µM) for 2 h. Then, the relative NAD^+^ and ATP levels were monitored. Results were separately repeated at least three times in triplicate. (**I**,**J**) Same cells as those in Figure 2C were treated with DMSO or IP-DNQ (0.05, 0.1, 0.15, 0.2, or 0.25 µM) for 2 h. Then, the relative NAD^+^ and ATP levels were monitored. Results were separately repeated at least three times in triplicate. (**K**,**L**) Same cells as those in Figure 2C were treated with DMSO or IP-DNQ (0.25 µM) for different time points (0 -120 min). Cells were collected and assessed for the relative NAD^+^ and ATP levels. For panels (**A**–**C**) and (**I**–**L**), each group had three replicates for every biological repetition. All error bars (mean ± SD) are derived from three independent experiments. *** *p* < 0.001, * *p* < 0.05 (*t* tests). The uncropped blots are shown in Appendix A.

**Figure 3 cancers-15-05844-f003:**
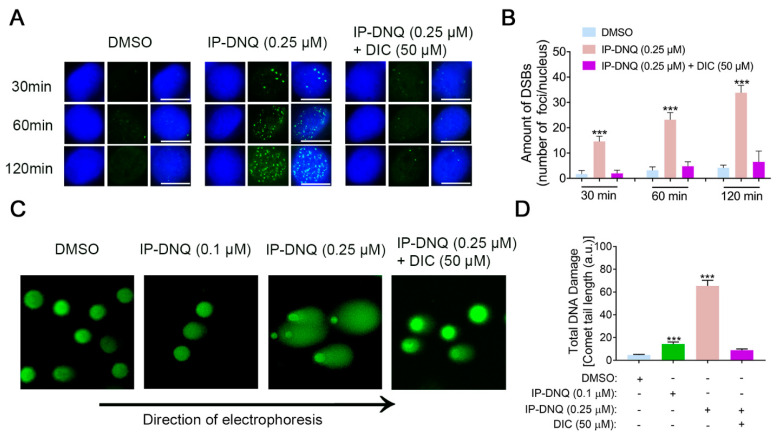
**NQO1 is required for IP-DNQ-mediated DNA damage in A549 cells.** (**A**) Representative images of A549 cells exposed to DMSO, IP-DNQ (0.25 µM), or IP-DNQ + DIC (50 µM); cells were assessed for DSB breaks over time (30, 60, and 120 min) using γ-H2AX as the surrogate marker (in green). Cells were also stained for nuclear DNA using DAPI (in blue). Scale bar, 10 μm. (**B**) Graphical representation of data presented in Figure 4A. (**C**) Total DNA damage by alkaline comet assay in A549 cells after IP-DNQ (0.1 or 0.25 µM) ± DIC (50 µM) for 2 h. (**D**) Comet tail lengths (arbitrary units) were assessed using NIH Image J software. Points or means were counted from 100 cells per treatment group. Each group had three replicates for every biological repetition. Results (mean ± SD) were derived from three independent experiments. *** *p* < 0.001 (*t* tests).

**Figure 4 cancers-15-05844-f004:**
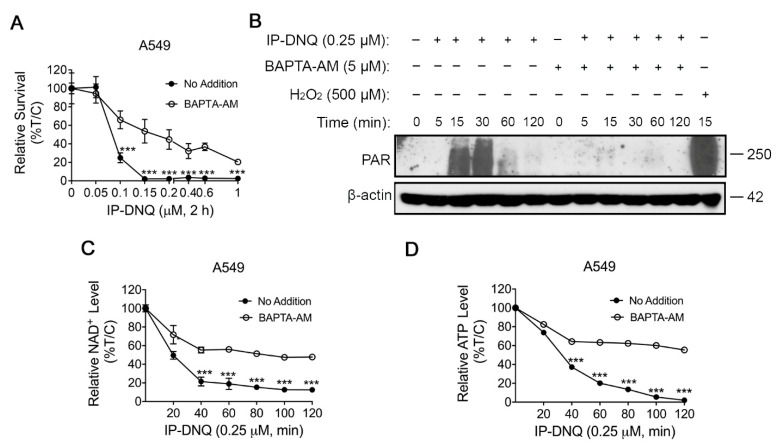
**Ca^2+^ plays a pivotal role in IP-DNQ-induced cytotoxicity.** (**A**) Long-term survival of A549 cells pretreated with or without BAPTA-AM (5 µM, 60 min), then cotreated with IP-DNQ (0–1 µM) for 2 h. (**B**) Western blot assays confirmed PAR formation in A549 cells pretreated with or without BAPTA-AM (5 µM, 60 min) ± IP-DNQ (0.25 µM) at indicated times (0–120 min). Cells were treated with H_2_O_2_ (500 µM, 15 min in PBS) as a positive control. Loading was controlled by β-actin levels. (**C**,**D**) A549 cells pretreated with or without BAPTA-AM (5 µM, 60 min) for 1 h, then cotreated with IP-DNQ (0–1 µM) at the indicated time points; cells were assessed for NAD^+^ (**C**) and ATP (**D**) levels. For panel A, each group had six replicates for every biological repetition; for panels C-D, each group had three replicates for every biological repetition. Results (mean ± SD) were derived from three independent experiments. *** *p* < 0.001 (*t* tests). The uncropped blots are shown in Appendix A.

**Figure 5 cancers-15-05844-f005:**
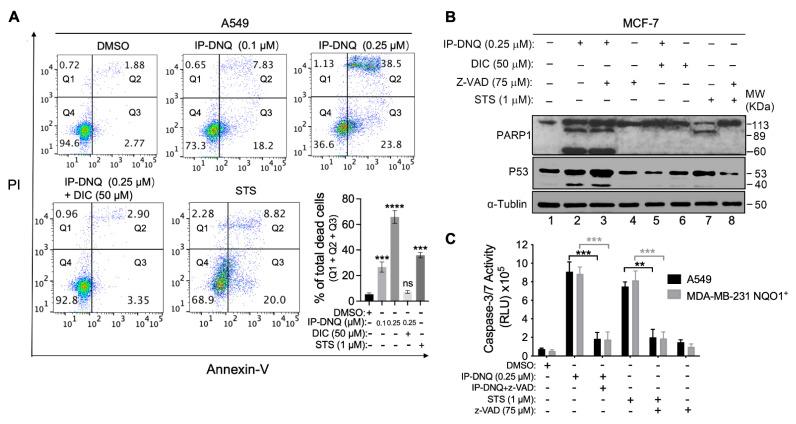
**IP-DNQ causes NQO1-dependent apoptosis and necrosis.** (**A**) A549 cells were treated with DMSO or IP-DNQ (0.1 or 0.25 µM) ± DIC (50 µM) for 2 h. Cells were also treated with 1 µM staurosporine (STS) for 18 h as an apoptotic positive control. Cells were collected after 48 h and assessed by flow cytometry. (**B**) MCF-7 cells were treated with IP-DNQ (0.25 μM) ± ZVAD-fmk (pan-caspase inhibitor, 75 μM) ± DIC (50 μM) for 2 h. Cells were also treated with 1 μM STS for 18 h or STS (1 μM, 18 h) ± ZVAD-fmk (75 μM, 2 h) to detect cell death pathways. The cells were collected after 24 h, and Western blot analysis of PARP1, p53, or α-tubulin for loading control was conducted. (**C**) A549 or MDA-MB-231 *NQO1^+^* cells were treated as described in (**B**), and cells were monitored for caspase-3/7 activation after 48 h. For panels A and C, each group had three replicates for every biological repetition. Results (mean ± SD) were derived from three independent experiments. **** *p* < 0.0001, *** *p* < 0.001, ** *p* < 0.01, and ns, not significant (*t* tests). The uncropped blots are shown in Appendix A.

**Figure 6 cancers-15-05844-f006:**
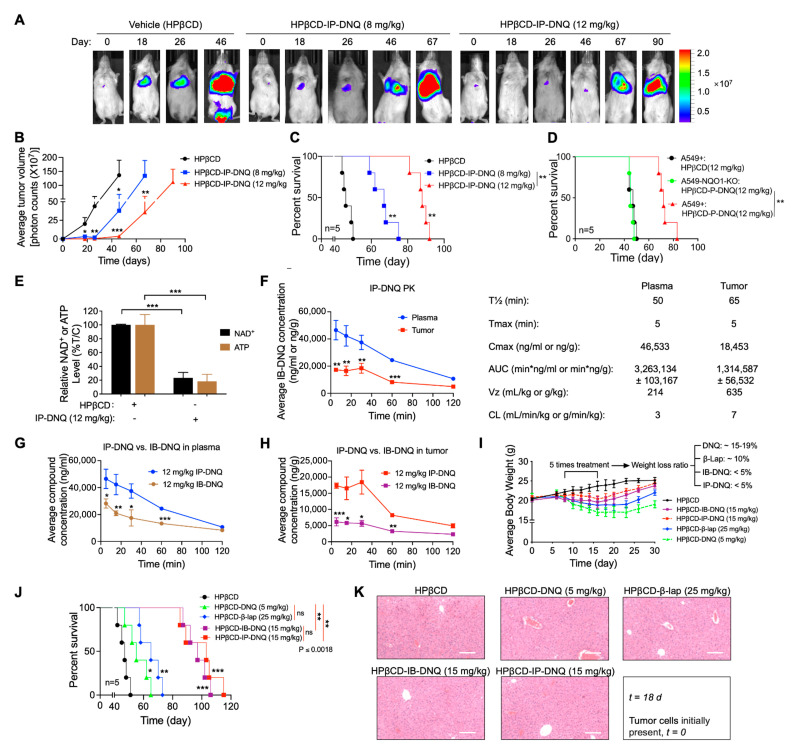
**IP-DNQ elicits antitumor efficacy against A549 NSCLC orthotopic Xenografts.** (**A**–**D**) A549 and *NQO1* knockout (A549-*NQO1*-KO) orthotopic lung tumors were established in 18–20 g female NSG mice (n = 5/group) by injecting ~1 × 10^6^ cells (tail vein, iv). After seven days, mice were treated with vehicle (HPβCD, iv), or HPβCD-IP-DNQ (8 or 12 mg/kg, iv) every other day for a total of five treatments. Bioluminescence imaging (BLI) monitored relative tumor volumes. (**A**) Representative BLI images from days 0, 18, 26, 46 67, and 90 are displayed. (**B**) Average tumor volumes were quantified on the days indicated in Figure 6A. (**C**) Kaplan–Meier survival curves for mice with A549 orthotopic tumors. (**D**) Kaplan–Meier survival curves for mice with A549 *NQO1*-KO orthotopic tumors. (**E**–**G**) Orthotopic A549 tumor-bearing female NSG mice (n = 3/group) were treated as described in (**A**–**D**), but with only one dosing, and they were sacrificed at the given time points. (**E**) Two hours post-drug treatment, tumor tissues were harvested and analyzed for relative NAD^+^ and ATP levels. (**F**,**G**) Pharmacokinetics (PK) of IP-DNQ/IB-DNQ were evaluated. (**I**–**K**) Orthotopic A549 tumor-bearing female NSG mice (n = 8/group) were treated as outlined in A-D, using the MTD doses of HPβCD-IP-DNQ (15 mg/kg), HPβCD-IB-DNQ (15 mg/kg), HPβCD-DNQ (5 mg/kg), or HPβCD-β-lap (25 mg/kg). (**I**) Mice body weights were tracked for a total of 30 days post tumor cell inoculation. (**J**) Kaplan–Meier survival curves for mice with A549 orthotopic tumors. (**K**) On day 18, three mice from each group were sacrificed for HE staining of the liver, scale bar = 10 μm. Results (mean ± SD) were derived from three independent experiments. For panels (**B**,**E**–**H**), *** *p* < 0.001, ** *p* < 0.01, and * *p* < 0.05 (*t* tests); for panels (**C**,**D**,**J**), *** *p <* 0.001, ** *p <* 0.01, * *p <* 0.05, and ns, not significant (log-rank test).

## Data Availability

Data available upon request.

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
