# Peer review of "Augmented Concentration of Isopentyl-Deoxynyboquinone in Tumors Selectively Kills NAD(P)H Quinone Oxidoreductase 1-Positive Cancer Cells through Programmed Necrotic and Apoptotic Mechanisms"

_cancers, 2023, doi:10.3390/cancers15245844_

Round 1

Reviewer 1 Report (Previous Reviewer 1)

Comments and Suggestions for Authors

The manuscript by Wang et al has been significantly improved with additional experiments to support the conclusions of the study. While the mechanism of DNQ-mediated cytotoxicity is not new, this manuscript presents a modification that potentially reduces the toxicity issues that plague the clinical use of previous compounds. This in itself may be of interest to our clinical colleagues. However, some issues still needs to be addressed before I can recommend this manuscript for publication.

Major comment:

In figure 5B why is MCF-7 used in this panel? All data in this manuscript used A549 lung cancer as a model for mechanism of action studies. The use of MCF-7 only in this panel raises some doubts about the validity of this figure. Is PARP cleavage not seen in A549? 

Comments on the Quality of English Language

The writing has been significantly improved. However, some minor edits may still be required for grammar and clarity. 

Typo in Pg 2 line 85, repeated sentence.

Author Response

We sincerely appreciate the reviewer's positive comments on our work and his/her valuable feedback. We have attached the point-by-point responses.

Reviewer 1:

Comments and Suggestions for Authors

The manuscript by Wang et al has been significantly improved with additional experiments to support the conclusions of the study. While the mechanism of DNQ-mediated cytotoxicity is not new, this manuscript presents a modification that potentially reduces the toxicity issues that plague the clinical use of previous compounds. This in itself may be of interest to our clinical colleagues. However, some issues still need to be addressed before I can recommend this manuscript for publication.

Response: We sincerely appreciate the reviewer's positive comments on our work and his/her valuable feedback.

Major comment:

In figure 5B why is MCF-7 used in this panel? All data in this manuscript used A549 lung cancer as a model for mechanism of action studies. The use of MCF-7 only in this panel raises some doubts about the validity of this figure. Is PARP cleavage not seen in A549? 

Response: PARP cleavage is indeed observed in A549 cells. However, nonspecific blots (red color) were consistently detected in this cell line, which is why we did not include this data in our figures (see attached PDF file).

Comments on the Quality of English Language

The writing has been significantly improved. However, some minor edits may still be required for grammar and clarity. 

Typo in Pg 2 line 85, repeated sentence.

Response: We apologize for the mistake. We have deleted the repeated sentence.

Reviewer 2 Report (Previous Reviewer 3)

Comments and Suggestions for Authors

Thank you to the authors for their efforts in responding to all of my questions and suggestions. The manuscript is much improved. I only have a few more comments before this manuscript can be accepted for publication. Most of my suggestions focus on more toxicity data as this is a pivotal part for this study. Please see my comments (in blue) to your responses (in black) in the text below.

Response: We appreciate the insightful question raised by the reviewer. PARylation is a post-translational protein modification mediated by poly(ADP-ribose) polymerase (PARP) enzymes. These enzymes utilize NAD+ as a substrate to add ADP-ribose polymers (PAR) to target proteins, regulating their function. PARylation primarily occurs at or near sites of DNA damage, facilitating the recruitment of DNA repair factors through their poly ADP-ribose (PAR) binding domains.

In Fig 2E, we observed that exposure of A549 cells to sublethal doses (0.1 μM) and lethal doses (0.2 μM) of IP-DNQ for 5 minutes resulted in the induction of increasing DNA damage, leading to a corresponding increase in PAR formation. However, at super-lethal doses of IP-DNQ (0.4, 0.6, and 0.8 μM), the consumption of NAD+ was rapid and exceeded the available supply within 5 minutes. As a result, PAR formation exhibited a decreasing trend under these conditions.

We believe this phenomenon can be attributed to the depletion of NAD+ by the higher doses of IP-DNQ, which limits the availability of substrate for PARP-mediated PARylation. Consequently, the concentration-dependent increase in PAR levels observed at lower IP-DNQ doses transitions to a decreased trend at super-lethal doses due to NAD+ depletion.

If this is the case, then Fig. 2E should be replaced with IP-DNQ treatment at 10, 20, 50, 80 and 100 nM instead of the concentrations used currently.

Response: We appreciate the reviewer's question. In our study, we did observe differences in tumor growth rates between WT A549 cells and A549-NQO1-KO cells. The A549-NQO1-KO cells exhibited a slower tumor growth compared to the WT cells. To compensate for the different tumor growth rates, we adjusted the initial cell numbers implanted for each cell line to ensure comparable tumor sizes at the start of the experiment. By carefully controlling the initial cell numbers, we aimed to minimize the impact of different growth rates on the subsequent analyses and comparisons of treatment effects between the two cell lines.

Could you please add these data as a supplemental figure comparing WT A549 cells and A549-NQO1-KO cells and also indicate in the methods what cell numbers were used to make this compensation and include data showing that the compensation resulted in similar tumor growth when untreated.

Response: We appreciate the reviewer's suggestion. In addition to the in vivo studies conducted using the A549 xenograft model, we have also performed experiments in a pancreatic xenograft model to evaluate the efficacy of IP-DNQ. These studies demonstrated potent antitumor efficacy in the pancreatic xenograft model as well. However, the results from the pancreatic xenograft model are part of a separate project focusing on IP-DNQ-mediated mitochondrial dysfunction in pancreatic cancer cells. Therefore, we did not include those data in the current manuscript. We believe that the inclusion of the additional xenograft model data would significantly enhance the scope and impact of our study. We plan to incorporate these findings in a separate publication, highlighting the broader therapeutic potential of IP-DNQ in different cancer types.

Please include these data on pancreatic cancer model in the response which is not published. As reviewers, we need to make sure of the translatability of your data across different models.

Response: We appreciate the reviewer's insightful question. We apologize for the misleading information in Fig. 6a and 6c. According to our previous studies (Huang et al, Cancer Cell, 2016 & Jiang et al, Frontiers in Oncology), A549 cells were injected into mice via tail vein injection, and we monitored the occurrence of BLI images. We observed BLI images approximately 4 hours after cell inoculation, indicating the accumulation of A549-Luciferase cells in the lungs after blood circulation. However, these images would disappear as the majority of the cancer cells were eliminated by the mice. Since NSG/NOD-SCID mice are deficient only in T cells and B cells, they still possess the ability to eliminate exogenous cancer cells. The first detectable BLI images surface around 18-19 days post A549 cell injection, with none being observable on day 0.

For clarity, we've updated Fig. 6A to feature the BLI image from each group on day 0, from the IP-DNQ (8 mg/kg) group on day 67, and from the IP-DNQ (12 mg/kg) group on days 18, 67, and 90. Additionally, we've rearranged the data in Fig. 6B. Notably, post day 46, members of the control group began to succumb,

Please include these BLI images for 4 hrs post cell inoculation in the response to reviewer’s comments to make sure similar numbers of cells were injected in all mice across different treatment groups.

Response: We appreciate the reviewer's suggestion. We have included the HE staining of liver for drug toxicity assay (Fig. 6K). We have also included body weight loss after 5 doses of drug administration to monitor the drug toxicity for mice (Fig. 6I).

Though HE images of livers can grossly show any abnormalities, it does not pick up actual molecular changes. I suggest the authors look at circulating levels of AST, ALT, alkaline phosphatase, bilirubin, and GGT in the blood of mice treated with at least IP-DNQ and DNQ (if possible, with all drugs used). DNQ showed a significant decrease in body weight which may go hand in hand with liver toxicity (not detected grossly by HE in Fig. 6k but may be detected molecularly).

Also, in the discussion the authors mentioned: “Our previous studies have demonstrated 514 β-lap and DNQ treatments induce methemoglobinemia”. It would be good to add levels of methemoglobinmia induced by IB- or IP-DNQ to clearly show that this new compounds has a better clinical profile with less side effects.

Reviewer: 14. Discussion on fig 1: “IP-DNQ could achieve efficacious potency using lower dose compared with β-lap in NQO1+-overexpressing MCF breast and A549 NSCL cancers (Fig. 1A, B, E, F).” Where is the data on Beta-lap and DNQ? Response: We apologize for the missing leading. We have rewritten this part in the discussion.

Please state clearly in the response what changes were made in the discussion to address my comment.

Reviewer: 16. Line 507: “In summary, our study provides strong preclinical evidence that IP-DNQ is a novel 507 and potent NQO1-dependent antitumor drug to induce multiple cell death pathways such 508 as apoptosis and necrosis. In addition, it possesses greater anti-tumor efficacy and best 509 biosafety profiles compared with any other NQO1 bioactivatable drugs that we tested be-510 fore. We believe that IP-DNQ can be used in clinical trials to benefit those patients with 511 NQO1-positive tumors and become a promising targeted therapeutic drug in oncology 512 field.” I do not think that given the data provide the authors can make these strong conclusions. The study is limited by the use of only one in vitro and in vivo model and to make these broad conclusions, more data is necessary. Please tone down the conclusions since they only apply to the A549 model (according to current data).

Response: We apologize for the missing leading. We have rewritten this part in the discussion.

Please state clearly in the response what changes were made in the discussion to address my comment.

Reviewer: 2. Line 501: “methemoglobin was measured following a previous published method [40], 501 and we could see the IP-DNQ produced minimal methemoglobin (-2.5%) compared with 502 DNQ, which treatment in DNQ mice caused severe methemoglobin symptoms (2.9%).” Please include these data in the manuscript.

Response: We apologize for the missing leading. We have rewritten this part in the discussion.

These data are still missing. Please add these data as also requested in my fifth response when discussion toxicity data.

Comments on the Quality of English Language

English language and style are fine/minor spell check required.

Author Response

Please find attached the point-by-point responses to the reviewer.

Round 2

Reviewer 1 Report (Previous Reviewer 1)

Comments and Suggestions for Authors

The authors have sufficiently address my concerns.

This manuscript is a resubmission of an earlier submission. The following is a list of the peer review reports and author responses from that submission.

Round 1

Reviewer 1 Report

Comments and Suggestions for Authors

Please see attached word file for comments.

Reviewer 2 Report

Comments and Suggestions for Authors

The study by Wang et al. is conceived and presented in a very poor manner.  In addition, most of the sections in the text are incomprehensible, as they are redacted in an equally poor manner and the english language falls below any acceptable standard.  Finally most of the figures are structured very poorly.  Overall, the content and the presentation of the data fall below any acceptable standard for consideration in this or any other journal.   

Reviewer 3 Report

Comments and Suggestions for Authors

The manuscript by Wang et. A. titled “Augmented Concentration of IP-DNQ in Tumors Selectively Kills NQO1-positive Cancer Cells by Programmed Necrotic and Apoptotic Mechanisms” presents a novel chemotherapeutic agent called IP-DNQ which induces ROS generation, DNA damage, and PARP1 hyperactivation resulting in cell death via apoptosis and necrosis pathways. This agent also performed very well in vivo reducing tumor growth and extending mouse survival. Even though the data is convincing, the data presented is limited by only one cell line for in vitro studies and only one xenograft mouse model for in vivo studies. Other in vitro and in vivo models of lung/breast/colon cancers are necessary to convincingly suggest that IP-DNQ kills cancer cells. Furthermore, in vivo efficacy studies are not accompanied by toxicity studies to show that IP-DNQ does not induce adverse effects in mice.

Major points:

1.     Please include a graph with NQ01 levels in the cell lines used.

2.     The structure of compound IP-DNQ should be included if it is a new compound (unless there is IP issues). If it was reported in another manuscript, the authors should still include the visual structure and compare it to beta-lap and DNQ.

3.     Fig 2E: why is there not an IP-DNQ-concentration dependent increase in PAR levels? Please explain

4.     Why was A549 chosen as the cell line for subsequent studies?

5.     Fig. 4 shows that Ca2+ is partially involved in IP-DNQ-induced cytotoxicity. Please elaborate on other possible mechanisms that would explain why survival is not completely rescued after addition of BAPTA-AM

6.     The data in A549 cells is convincing; however, with the data show the readers cannot tell whether the effect is something specific to A549 cells or whether the induction of DNA double breaks and PARP1 overactivation is broad to other NQ01 overexpressing cancer cells. Please add 2-3 more cells lines for key experiments like those in FIg. 5.

7.     Is it know how IP-DNQ binds to NQ01? If so, could you please add in silico projected 3D structures of IP-DNQ bound to NQ01 and if possible, how DIC competes for binding to NQ01?

8.     Are there differences in the tumor growth rates of WT A549 cells and A549-NQ01-KO cells? If so, how did you compensate for the different tumor growth rates?

9.     As stated in a previous comment, the authors only used one in vivo model (A549 cells in NSG mice) to test the efficacy of IP-DNQ. It is important to include experiments in another xenograft model of lung cancer or in another cancer type.

10.  Fig. 6a and 6c: By BLI there is a pronounced decrease in tumor growth rates in the 12 mg/kg IP-DNQ group compared to the vehicle and yet these mice died in average by days 90-95 according to the KM survival plot. The authors should include BLI images for IP-DNQ groups at days 0-7 and 90-95. Otherwise, the data in Fig. 6a is misleading.

11.  Fig. 6A: authors need to include a day 0 and 7 BLI images for each treatment group to determine what the size of the tumors were before treatment started.

12.  For the in vivo studies, the authors should add treatment data with the most promising NQ01 inhibitor currently available to clearly show that IP-DNQ has a therapeutic advantage over the most potent current inhibitor or at least comparisons to beta-lap and DNQ in NSG mice.

13.  The in vivo data is lacking toxicity studies. There is no data on the in liver and kidney function studies and body weights OVER TIME. According to the survival graphs mice were treated 5 times every other day and then maintained for up to 90-95 days (in the 12 mg/kg IP-DNQ group). There should be toxicity data at different time points in this 90 day period.

14.  Discussion on fig 1: “IP-DNQ could achieve efficacious potency using lower dose compared with β-lap in NQO1+-overexpressing MCF breast and A549 NSCL cancers (Fig. 1A, B, E, F).” Where is the data on Beta-lap and DNQ?

15.  In Supplemental table 2 the authors describe the effects of the compounds used as “moderate”, “severe” or “less”. The use of this words is not quantitative. Please use a clinical scale method to score the severity of tachypnea and jumping and hunched backs in these mice.

16.  Line 507: “In summary, our study provides strong preclinical evidence that IP-DNQ is a novel 507 and potent NQO1-dependent antitumor drug to induce multiple cell death pathways such 508 as apoptosis and necrosis. In addition, it possesses greater anti-tumor efficacy and best 509 biosafety profiles compared with any other NQO1 bioactivatable drugs that we tested be-510 fore. We believe that IP-DNQ can be used in clinical trials to benefit those patients with 511 NQO1-positive tumors and become a promising targeted therapeutic drug in oncology 512 field.” I do not think that given the data provide the authors can make these strong conclusions. The study is limited by the use of only one in vitro and in vivo model and to make this broad conclusions, more data is necessary. Please tone down the conclusions since they only apply to the A549 model (according to current data).

17.  The discussion needs to be re-written completely. As of right now, this section only reiterates what was shown in the results section. There is no actual discussion of how IP-DNQ may behave clinically, study limitations, clinical expectations of this drug, and how IP-DNQ fits in the lung cancer field. The discussion section is not a reiteration of the results but a section to interpret and explain unexpected results and relate to what others have done in the field. 

Minor points:

1.     The authors should state in the figure legends the number of replicates for each treatment group used in each experiment. The size of some figures (e.g. fig. 5) is very small and need to be reformatted.

2.     Line 501: “methemoglobin was measured following a previous published method [40], 501 and we could see the IP-DNQ produced minimal methemoglobin (-2.5%) compared with 502 DNQ, which treatment in DNQ mice caused severe methemoglobin symptoms (2.9%).” Please include these data in the manuscript.

3.     Fig.1: change the labels on the treatment groups. “No addition” sounds like nothing was added but if I understand correctly, these treatment groups was treated with IP-DNQ.

4.     Correct grammatical errors and typos

5.     Line 464: “the side effects were significant” List what the side effects were.